# Learning Prices for Repeated Auctions with Strategic Buyers

**Kareem Amin**
University of Pennsylvania
akareem@cis.upenn.edu

**Afshin Rostamizadeh**
Google Research
rostami@google.com

**Umar Syed**
Google Research
usyed@google.com

## Abstract

Inspired by real-time ad exchanges for online display advertising, we consider the problem of inferring a buyer's value distribution for a good when the buyer is repeatedly interacting with a seller through a posted-price mechanism. We model the buyer as a strategic agent, whose goal is to maximize her long-term surplus, and we are interested in mechanisms that maximize the seller's long-term revenue. We define the natural notion of *strategic regret* — the lost revenue as measured against a truthful (non-strategic) buyer. We present seller algorithms that are no-(strategic)-regret when the buyer discounts her future surplus — i.e. the buyer prefers showing advertisements to users sooner rather than later. We also give a lower bound on strategic regret that increases as the buyer's discounting weakens and shows, in particular, that any seller algorithm will suffer linear strategic regret if there is no discounting.

## 1   Introduction

Online display advertising inventory — e.g., space for banner ads on web pages — is often sold via automated transactions on real-time ad exchanges. When a user visits a web page whose advertising inventory is managed by an ad exchange, a description of the web page, the user, and other relevant properties of the *impression*, along with a *reserve price* for the impression, is transmitted to bidding servers operating on behalf of advertisers. These servers process the data about the impression and respond to the exchange with a bid. The highest bidder wins the right to display an advertisement on the web page to the user, provided that the bid is above the reserve price. The amount charged the winner, if there is one, is settled according to a second-price auction. The winner is charged the maximum of the second-highest bid and the reserve price.

Ad exchanges have been a boon for advertisers, since rich and real-time data about impressions allow them to target their bids to only those impressions that they value. However, this precise targeting has an unfortunate side effect for web page publishers. A nontrivial fraction of ad exchange auctions involve only a *single* bidder. Without competitive pressure from other bidders, the task of maximizing the publisher's revenue falls entirely to the reserve price setting mechanism. Second-price auctions with a single bidder are equivalent to *posted-price* auctions. The seller offers a price for a good, and a buyer decides whether to accept or reject the price (i.e., whether to bid above or below the reserve price).

In this paper, we consider online learning algorithms for setting prices in posted-price auctions where the seller repeatedly interacts with the *same* buyer over a number of rounds, a common occurrence in ad exchanges where the same buyer might be interested in buying thousands of user impressions daily. In each round $t$, the seller offers a good to a buyer for price $p_t$. The buyer's value $v_t$ for the good is drawn independently from a fixed value distribution. Both $v_t$ and the value distribution are known to the buyer, but neither is observed by the seller. If the buyer accepts price $p_t$, the seller receives revenue $p_t$, and the buyer receives *surplus* $v_t - p_t$. Since the same buyer participates in

the auction in each round, the seller has the opportunity to *learn* about the buyer's value distribution and set prices accordingly. Notice that in worst-case repeated auctions there is no such opportunity to learn, while standard Bayesian auctions assume knowledge of a value distribution, but avoid addressing how or why the auctioneer was ever able to estimate this distribution.

Taken as an online learning problem, we can view this as a 'bandit' problem [18, 16], since the revenue for any price not offered is not observed (e.g., even if a buyer rejects a price, she may well have accepted a lower price). The seller's goal is to maximize his expected revenue over all $T$ rounds. One straightforward way for the seller to set prices would therefore be to use a *no-regret* bandit algorithm, which minimizes the difference between seller's revenue and the revenue that would have been earned by offering the best fixed price $p^*$ in hindsight for all $T$ rounds; for a no-regret algorithm (such as UCB [3] or EXP3 [4]), this difference is $o(T)$. However, we argue that traditional no-regret algorithms are inadequate for this problem. Consider the motivations of a buyer interacting with an ad exchange where the prices are set by a no-regret algorithm, and suppose for simplicity that the buyer has a fixed value $v_t = v$ for all $t$. The goal of the buyer is to acquire the most valuable advertising inventory for the least total cost, i.e., to maximize her total surplus $\sum_t v - p_t$, where the sum is over rounds where the buyer accepts the seller's price. A naive buyer might simply accept the seller's price $p_t$ if and only if $v_t \geq p_t$; a buyer who behaves this way is called *truthful*. Against a truthful buyer any no-regret algorithm will eventually learn to offer prices $p_t \approx v$ on nearly all rounds. But a more savvy buyer will notice that if she rejects prices in earlier rounds, then she will tend to see lower prices in later rounds. Indeed, suppose the buyer only accepts prices below some small amount $\epsilon$. Then any no-regret algorithm will learn that offering prices above $\epsilon$ results in zero revenue, and will eventually offer prices below that threshold on nearly all rounds. In fact, the smaller the learner's regret, the faster this convergence occurs. If $v \gg \epsilon$ then the deceptive buyer strategy results in a large gain in total surplus for the buyer, and a large loss in total revenue for the seller, relative to the truthful buyer. While the no-regret guarantee certainly holds — in hindsight, the best price is indeed $\epsilon$ — it seems fairly useless.

In this paper, we propose a definition of *strategic regret* that accounts for the buyer's incentives, and give algorithms that are no-regret with respect to this definition. In our setting, the seller chooses a learning algorithm for selecting prices and announces this algorithm to the buyer. We assume that the buyer will examine this algorithm and adopt whatever strategy maximizes her expected surplus over all $T$ rounds. We define the seller's strategic regret to be the difference between his expected revenue and the expected revenue he would have earned if, rather than using his chosen algorithm to set prices, he had instead offered the best fixed price $p^*$ on all rounds *and the buyer had been truthful*. As we have seen, this revenue can be much higher than the revenue of the best fixed price in hindsight (in the example above, $p^* = v$). Unless noted otherwise, throughout the remainder of the paper the term "regret" will refer to strategic regret.

We make one further assumption about buyer behavior, which is based on the observation that in many important real-world markets — and particularly in online advertising — sellers are far more willing to wait for revenue than buyers are willing to wait for goods. For example, advertisers are often interested in showing ads to users who have recently viewed their products online (this practice is called 'retargeting'), and the value of these user impressions decays rapidly over time. Or consider an advertising campaign that is tied to a product launch. A user impression that is purchased long after the launch (such as the release of a movie) is almost worthless. To model this phenomenon we multiply the buyer's surplus in each round by a *discount factor*: If the buyer accepts the seller's price $p_t$ in round $t$, she receives surplus $\gamma_t(v_t - p_t)$, where $\{\gamma_t\}$ is a nonincreasing sequence contained in the interval $(0, 1]$. We call $T_\gamma = \sum_{t=1}^{T} \gamma_t$ the buyer's 'horizon', since it is analogous to the seller's horizon $T$. The buyer's horizon plays a central role in our analysis.

**Summary of results:** In Sections 4 and 5 we assume that discount rates decrease geometrically: $\gamma_t = \gamma^{t-1}$ for some $\gamma \in (0, 1]$. In Section 4 we consider the special case that the buyer has a fixed value $v_t = v$ for all rounds $t$, and give an algorithm with regret at most $O(T_\gamma \sqrt{T})$. In Section 5 we allow the $v_t$ to be drawn from any distribution that satisfies a certain smoothness assumption, and give an algorithm with regret at most $\tilde{O}(T^\alpha + T_\gamma^{1/\alpha})$ where $\alpha \in (0, 1)$ is a user-selected parameter. Note that for either algorithm to be no-regret (i.e., for regret to be $o(T)$), we need that $T_\gamma = o(T)$. In Section 6 we prove that this requirement is necessary for no-regret: any seller algorithm has regret at least $\Omega(T_\gamma)$. The lower bound is proved via a reduction to a non-repeated, or 'single-shot', auction. That our regret bounds should depend so crucially on $T_\gamma$ is foreshadowed by the example above, in

which a deceptive buyer foregoes surplus in early rounds to obtain even more surplus is later rounds. A buyer with a short horizon $T_\gamma$ will be unable to execute this strategy, as she will not be capable of bearing the short-term costs required to manipulate the seller.

## 2 Related work

Kleinberg and Leighton study a posted price repeated auction with goods sold sequentially to $T$ bidders who either all have the same fixed private value, private values drawn from a fixed distribution, or private values that are chosen by an oblivious adversary (an adversary that acts independently of observed seller behavior) [15] (see also [7, 8, 14]). Cesa-Bianchi et al. study a related problem of setting the reserve price in a second price auction with multiple (but not repeated) bidders at each round [9]. Note that none of these previous works allow for the possibility of a strategic buyer, i.e. one that acts non-truthfully in order to maximize its surplus. This is because a new buyer is considered at each time step and if the seller behavior depends only on previous buyers, then the setting immediately becomes *strategyproof*.

Contrary to what is studied in these previous theoretical settings, electronic exchanges in practice see the same buyer appearing in multiple auctions and, thus, the buyer has incentive to act strategically. In fact, [12] finds empirical evidence of buyers' strategic behavior in sponsored search auctions, which in turn negatively affects the seller's revenue. In the economics literature, 'intertemporal price discrimination' refers to the practice of using a buyer's past purchasing behavior to set future prices. Previous work [1, 13] has shown, as we do in Section 6, that a seller cannot benefit from conditioning prices on past behavior if the buyer is not myopic and can respond strategically. However, in contrast to our work, these results assume that the seller knows the buyer's value distribution.

Our setting can be modeled as a nonzero sum repeated game of incomplete information, and there is extensive literature on this topic. However, most previous work has focused only on characterizing the equilibria of these games. Further, our game has a particular structure that allows us to design seller algorithms that are much more efficient than generic algorithms for solving repeated games.

Two settings that are distinct from what we consider in this paper, but where mechanism design and learning are combined, are the multi-armed bandit mechanism design problem [6, 5, 11] and the incentive compatible regression/classification problem [10, 17]. The former problem is motivated by sponsored search auctions, where the challenge is to elicit truthful values from multiple bidding advertisers while also efficiently estimating the click-through rate of the set of ads that are to be allocated. The latter problem involves learning a discriminative classifier or regression function in the batch setting with training examples that are labeled by selfish agents. The goal is then to minimize error with respect to the truthful labels.

Finally, Arora et al. proposed a notion of regret for online learning algorithms, called policy regret, that accounts for the possibility that the adversary may adapt to the learning algorithm's behavior [2]. This resembles the ability, in our setting, of a strategic buyer to adapt to the seller algorithm's behavior. However, even this stronger definition of regret is inadequate for our setting. This is because policy regret is equivalent to standard regret when the adversary is oblivious, and as we explained in the previous section, there is an oblivious buyer strategy such that the seller's standard regret is small, but his regret with respect to the best fixed price against a truthful buyer is large.

## 3 Preliminaries and Model

We consider a posted-price model for a single buyer repeatedly purchasing items from a single seller. Associated with the buyer is a fixed distribution $\mathcal{D}$ over the interval $[0, 1]$, which is known only to the buyer. On each round $t$, the buyer receives a value $v_t \in \mathcal{V} \subseteq [0, 1]$ from the distribution $\mathcal{D}$. The seller, without observing this value, then posts a price $p_t \in \mathcal{P} \subseteq [0, 1]$. Finally, the buyer selects an allocation decision $a_t \in \{0, 1\}$. On each round $t$, the buyer receives an *instantaneous surplus* of $a_t(v_t - p_t)$, and the seller receives an *instantaneous revenue* of $a_t p_t$.

We will be primarily interested in designing the seller's *learning algorithm*, which we will denote $\mathcal{A}$. Let $v_{1:t}$ denote the sequence of values observed on the first $t$ rounds, $(v_1, ..., v_t)$, defining $p_{1:t}$ and $a_{1:t}$ analogously. $\mathcal{A}$ is an algorithm that selects each price $p_t$ as a (possibly randomized) function of $(p_{1:t-1}, a_{1:t-1})$. As is common in mechanism design, we assume that the seller announces his

choice of algorithm $\mathcal{A}$ in advance. The buyer then selects her *allocation strategy* in response. The buyer's allocation strategy $\mathcal{B}$ generates allocation decisions $a_t$ as a (possibly randomized) function of $(\mathcal{D}, v_{1:t}, p_{1:t}, a_{1:t-1})$.

Notice that a choice of $\mathcal{A}$, $\mathcal{B}$ and $\mathcal{D}$ fixes a distribution over the sequences $a_{1:T}$ and $p_{1:T}$. This in turn defines the seller's total expected revenue:

$$\text{SellerRevenue}(\mathcal{A}, \mathcal{B}, \mathcal{D}, T) = E\left[\sum_{t=1}^{T} a_t p_t \mid \mathcal{A}, \mathcal{B}, \mathcal{D}\right].$$

In the most general setting, we will consider a buyer whose surplus may be discounted through time. In fact, our lower bounds will demonstrate that a sufficiently decaying discount rate is necessary for a no-regret learning algorithm. We will imagine therefore that there exists a nonincreasing sequence $\{\gamma_t \in (0, 1]\}$ for the buyer. For a choice of $T$, we will define the effective "time-horizon" for the buyer as $T_\gamma = \sum_{t=1}^{T} \gamma_t$. The buyer's expected total discounted surplus is given by:

$$\text{BuyerSurplus}(\mathcal{A}, \mathcal{B}, \mathcal{D}, T) = E\left[\sum_{t=1}^{T} \gamma_t a_t (v_t - p_t) \mid \mathcal{A}, \mathcal{B}, \mathcal{D}\right].$$

We assume that the seller is faced with a strategic buyer who adapts to the choice of $\mathcal{A}$. Thus, let $\mathcal{B}^*(\mathcal{A}, \mathcal{D})$ be a surplus-maximizing buyer for seller algorithm $\mathcal{A}$ and value distribution is $\mathcal{D}$. In other words, for all strategies $\mathcal{B}$ we have

$$\text{BuyerSurplus}(\mathcal{A}, \mathcal{B}^*(\mathcal{A}, \mathcal{D}), \mathcal{D}, T) \geq \text{BuyerSurplus}(\mathcal{A}, \mathcal{B}, \mathcal{D}, T).$$

We are now prepared to define the seller's regret. Let $p^* = \arg\max_{p \in \mathcal{P}} p \Pr_\mathcal{D}[v \geq p]$, the revenue-maximizing choice of price for a seller that *knows* the distribution $\mathcal{D}$, and simply posts a price of $p^*$ on every round. Against such a pricing strategy, it is in the buyer's best interest to be *truthful*, accepting if and only if $v_t \geq p^*$, and the seller would receive a revenue of $T p^* \Pr_{v \sim \mathcal{D}}[v \geq p^*]$. Informally, a no-regret algorithm is able to learn $\mathcal{D}$ from previous interactions with the buyer, and converge to selecting a price close to $p^*$. We therefore define regret as:

$$\text{Regret}(\mathcal{A}, \mathcal{D}, T) = T p^* \Pr_{v \sim \mathcal{D}}[v \geq p^*] - \text{SellerRevenue}(\mathcal{A}, \mathcal{B}^*(\mathcal{A}, \mathcal{D}), \mathcal{D}, T).$$

Finally, we will be interested in algorithms that attain $o(T)$ regret (meaning the averaged regret goes to zero as $T \to \infty$) for the worst-case $\mathcal{D}$. In other words, we say $\mathcal{A}$ is *no-regret* if $\sup_\mathcal{D} \text{Regret}(\mathcal{A}, \mathcal{D}, T) = o(T)$. Note that this definition of worst-case regret only assumes that Nature's behavior (i.e., the value distribution) is worst-case; the buyer's behavior is always presumed to be surplus maximizing.

## 4   Fixed Value Setting

In this section we consider the case of a single unknown fixed buyer value, that is $\mathcal{V} = \{v\}$ for some $v \in (0, 1]$. We show that in this setting a very simple pricing algorithm with monotonically decreasing price offerings is able to achieve $O(T_\gamma \sqrt{T})$ when the buyer discount is $\gamma_t = \gamma^{t-1}$. Due to space constraints many of the proofs for this section appear in Appendix A.

> Monotone algorithm: Choose parameter $\beta \in (0, 1)$, and initialize $a_0 = 1$ and $p_0 = 1$. In each round $t \geq 1$ let $p_t = \beta^{1 - a_{t-1}} p_{t-1}$.

In the `Monotone` algorithm, the seller starts at the maximum price of $1$, and decreases the price by a factor of $\beta$ whenever the buyer rejects the price, and otherwise leaves it unchanged. Since `Monotone` is deterministic and the buyer's value $v$ is fixed, the surplus-maximizing buyer algorithm $\mathcal{B}^*(\texttt{Monotone}, v)$ is characterized by a deterministic allocation sequence $a_{1:T}^* \in \{0, 1\}^T$.[1]

The following lemma partially characterizes the optimal buyer allocation sequence.

**Lemma 1.** *The sequence* $a_1^*, \dots, a_T^*$ *is monotonically nondecreasing.*

In other words, once a buyer decides to start accepting the offered price at a certain time step, she will keep accepting from that point on. The main idea behind the proof is to show that if there does exist some time step $t'$ where $a_{t'}^* = 1$ and $a_{t'+1}^* = 0$, then swapping the values so that $a_{t'}^* = 0$ and $a_{t'+1}^* = 1$ (as well potentially swapping another pair of values) will result in a sequence with strictly better surplus, thereby contradicting the optimality of $a_{1:T}^*$. The full proof is shown in Section A.1.

Now, to finish characterizing the optimal allocation sequence, we provide the following lemma, which describes time steps where the buyer has with certainty begun to accept the offered price.

**Lemma 2.** *Let* $c_{\beta,\gamma} = 1 + (1-\beta)T_\gamma$ *and* $d_{\beta,\gamma} = \frac{\log\left(\frac{c_{\beta,\gamma}}{v}\right)}{\log(1/\beta)}$, *then for any* $t > d_{\beta,\gamma}$ *we have* $a_{t+1}^* = 1$.

A detailed proof is presented in Section A.2. These lemmas imply the following regret bound.

**Theorem 1.** $\text{Regret}(\texttt{Monotone}, v, T) \leq vT \left(1 - \frac{\beta}{c_{\beta,\gamma}}\right) + v\beta \left(\frac{d_{\beta,\gamma}}{c_{\beta,\gamma}} + \frac{1}{c_{\beta,\gamma}}\right)$.

*Proof.* By Lemmas 1 and 2 we receive no revenue until at most round $\lceil d_{\beta,\gamma} \rceil + 1$, and from that round onwards we receive at least revenue $\beta^{\lceil d_{\beta,\gamma} \rceil}$ per round. Thus

$$\text{Regret}(\texttt{Monotone}, v, T) = vT - \sum_{t=\lceil d_{\beta,\gamma} \rceil + 1}^{T} \beta^{\lceil d_{\beta,\gamma} \rceil} \leq vT - (T - d_{\beta,\gamma} - 1)\beta^{d_{\beta,\gamma}+1}$$

Noting that $\beta^{d_{\beta,\gamma}} = \frac{v}{c_{\beta,\gamma}}$ and rearranging proves the theorem. $\square$

Tuning the learning parameter simplifies the bound further and provides a $O(T_\gamma\sqrt{T})$ regret bound. Note that this tuning parameter does not assume knowledge of the buyer's discount parameter $\gamma$.

**Corollary 1.** *If* $\beta = \frac{\sqrt{T}}{1+\sqrt{T}}$ *then* $\text{Regret}(\texttt{Monotone}, v, T) \leq \sqrt{T}\left(4vT_\gamma + 2v\log\left(\frac{1}{v}\right)\right) + v$.

The computation used to derive this corollary are found in Section A.3. This corollary shows that it is indeed possible to achieve no-regret against a strategic buyer with a unknown fixed value as long as $T_\gamma = o(\sqrt{T})$. That is, the effective buyer horizon must be more than a constant factor smaller than the square-root of the game's finite horizon.

## 5 Stochastic Value Setting

We next give a seller algorithm that attains no-regret when the set of prices $\mathcal{P}$ is finite, the buyer's discount is $\gamma_t = \gamma^{t-1}$, and the buyer's value $v_t$ for each round is drawn from a fixed distribution $\mathcal{D}$ that satistfies a certain continuity assumption, detailed below.

> Phased algorithm: Choose parameter $\alpha \in (0,1)$. Define $T_i \equiv 2^i$ and $S_i \equiv \min\left(\frac{T_i}{|\mathcal{P}|}, T_i^\alpha\right)$. For each phase $i = 1, 2, 3, \ldots$ of length $T_i$ rounds:
>
> Offer each price $p \in \mathcal{P}$ for $S_i$ rounds, in some fixed order; these are the *explore* rounds. Let $A_{p,i}$ = Number of explore rounds in phase $i$ where price $p$ was offered and the buyer accepted. For the remaining $T_i - |\mathcal{P}|S_i$ rounds of phase $i$, offer price $\tilde{p}_i = \arg\max_{p \in \mathcal{P}} p\frac{A_{p,i}}{S_i}$ in each round; these are the *exploit* rounds.

The `Phased` algorithm proceeds across a number of phases. Each phase consists of explore rounds followed by exploit rounds. During explore rounds, the algorithm selects each price in some fixed order. During exploit rounds, the algorithm repeatedly selects the price that realized the greatest revenue during the immediately preceding explore rounds.

First notice that a strategic buyer has no incentive to lie during exploit rounds (i.e. it will accept any price $p_t < v_t$ and reject any price $p_t > v_t$), since its decisions there do not affect any of its future prices. Thus, the exploit rounds are the time at which the seller can exploit what it has learned from the buyer during exploration. Alternatively, if the buyer has successfully manipulated the seller into offering a low price, we can view the buyer as "exploiting" the seller.

During explore rounds, on the other hand, the strategic buyer can benefit by telling lies which will cause it to witness better prices during the corresponding exploit rounds. However, the value of these lies to the buyer will depend on the fraction of the phase consisting of explore rounds. Taken to the extreme, if the entire phase consists of explore rounds, the buyer is not interested in lying. In general, the more explore rounds, the more revenue has to be sacrificed by a buyer that is lying during the explore rounds. For the myopic buyer, the loss of enough immediate revenue at some point ceases to justify her potential gains in the future exploit rounds.

Thus, while traditional algorithms like UCB balance exploration and exploitation to ensure confidence in the observed payoffs of sampled arms, our `Phased` algorithm explores for two purposes: to ensure accurate estimates, and to dampen the buyer's incentive to mislead the seller. The seller's balancing act is to explore for long enough to learn the buyer's value distribution, but leave enough exploit rounds to benefit from the knowledge.

**Continuity of the value distribution** The preceding argument required that the distribution $\mathcal{D}$ does not exhibit a certain pathology. There cannot be two prices $p, p'$ that are very close but $p \Pr_{v \sim \mathcal{D}}[v \geq p]$ and $p' \Pr_{v \sim \mathcal{D}}[v \geq p']$ are very different. Otherwise, the buyer is largely indifferent to being offered prices $p$ or $p'$, but distinguishing between the two prices is essential for the seller during exploit rounds. Thus, we assume that the value distribution $\mathcal{D}$ is $K$-*Lipschitz*, which eliminates this problem: Defining $F(p) \equiv \Pr_{v \sim \mathcal{D}}[v \geq p]$, we assume there exists $K > 0$ such that $|F(p) - F(p')| \leq K|p - p'|$ for all $p, p' \in [0, 1]$. This assumption is quite mild, as our `Phased` algorithm does not need to know $K$, and the dependence of the regret rate on $K$ will be logarithmic.

**Theorem 2.** *Assume $F(p) \equiv \Pr_{v \sim \mathcal{D}}[v \geq p]$ is K-Lipschitz. Let $\Delta = \min_{p \in \mathcal{P} \backslash \{p^*\}} p^* F(p^*) - pF(p)$, where $p^* = \arg\max_{p \in \mathcal{P}} pF(p)$. For any parameter $\alpha \in (0, 1)$ of the `Phased` algorithm there exist constants $c_1, c_2, c_3, c_4$ such that*

$$\mathrm{Regret}(\mathtt{Phased}, \mathcal{D}, T) \leq c_1 |\mathcal{P}| T^\alpha + c_2 \frac{|\mathcal{P}|}{\Delta^{2/\alpha}} (\log T)^{1/\alpha}$$

$$+ c_3 \frac{|\mathcal{P}|}{\Delta^{1/\alpha}} T_\gamma^{1/\alpha} (\log T + \log(K/\Delta))^{1/\alpha} + c_4 |\mathcal{P}|$$

$$= \tilde{O}(T^\alpha + T_\gamma^{1/\alpha}).$$

The complete proof of Theorem 2 is rather technical, and is provided in Appendix B.

To gain further intuition about the upper bounds proved in this section and the previous section, it helps to parametrize the buyer's horizon $T_\gamma$ as a function of $T$, e.g. $T_\gamma = T^c$ for $0 \leq c \leq 1$. Writing it in this fashion, we see that the `Monotone` algorithm has regret at most $O(T^{c+\frac{1}{2}})$, and the `Phased` algorithm has regret at most $\tilde{O}(T^{\sqrt{c}})$ if we choose $\alpha = \sqrt{c}$. The lower bound proved in the next section states that, in the worst case, any seller algorithm will incur a regret of at least $\Omega(T^c)$.

# 6   Lower Bound

In this section we state the main lower bound, which establishes a connection between the regret of any seller algorithm and the buyer's discounting. Specifically, we prove that the regret of any seller algorithm is $\Omega(T_\gamma)$. Note that when $T = T_\gamma$ — i.e., the buyer does not discount her future surplus — our lower bound proves that no-regret seller algorithms do not exist, and thus it is *impossible for the seller to take advantage of learned information*. For example, consider the seller algorithm that uniformly selects prices $p_t$ from $[0, 1]$. The optimal buyer algorithm is truthful, accepting if $p_t < v_t$, as the seller algorithm is non-adaptive, and the buyer does not gain any advantage by being more strategic. In such a scenario the seller would quickly learn a good estimate of the value distribution $\mathcal{D}$. What is surprising is that a seller cannot *use* this information if the buyer does not discount her future surplus. If the seller attempts to leverage information learned through interactions with the buyer, the buyer can react accordingly to negate this advantage.

The lower bound further relates regret in the repeated setting to regret in a particular single-shot game between the buyer and the seller. This demonstrates that, against a non-discounted buyer, the seller is no better off in the repeated setting than he would be by repeatedly implementing such a single-shot mechanism (ignoring previous interactions with the buyer). In the following section we describe the simple single-shot game.

## 6.1  Single-Shot Auction

We call the following game the *single-shot auction*. A seller selects a family of distributions $\mathcal{S}$ indexed by $b \in [0, 1]$, where each $\mathcal{S}_b$ is a distribution on $[0, 1] \times \{0, 1\}$. The family $\mathcal{S}$ is revealed to a buyer with unknown value $v \in [0, 1]$, who then must select a bid $b \in [0, 1]$, and then $(p, a) \sim \mathcal{S}_b$ is drawn from the corresponding distribution.

As usual, the buyer gets a surplus of $a(v - p)$, while the seller enjoys a revenue of $ap$. We restrict the set of seller strategies to distributions that are *incentive compatible* and *rational*. $\mathcal{S}$ is incentive compatible if for all $b, v \in [0, 1]$, $E_{(p,a)\sim\mathcal{S}_b}[a(v-p)] \leq E_{(p,a)\sim\mathcal{S}_v}[a(v-p)]$. It is *rational* if for all $v$, $E_{(p,a)\sim\mathcal{S}_v}[a(v-p)] \geq 0$ (i.e. any buyer maximizing expected surplus is actually incentivised to play the game). Incentive compatible and rational strategies exist: drawing $p$ from a fixed distribution (i.e. all $\mathcal{S}_b$ are the same), and letting $a = \mathbf{1}\{b \geq p\}$ suffices.[2]

We define the regret in the single-shot setting of any incentive-compatible and rational strategy $\mathcal{S}$ with respect to value $v$ as
$$\text{SSRegret}(\mathcal{S}, v) = v - E_{(p,a)\sim\mathcal{S}_v}[ap].$$

The following loose lower bound on $\text{SSRegret}(\mathcal{S}, v)$ is straightforward, and establishes that a seller's revenue cannot be a constant fraction of the buyer's value for all $v$. The full proof is provided in the appendix (Section C.1).

**Lemma 3.** *For any incentive compatible and rational strategy $\mathcal{S}$ there exists $v \in [0, 1]$ such that* $\text{SSRegret}(\mathcal{S}, v) \geq \frac{1}{12}$.

## 6.2  Repeated Auction

Returning to the repeated setting, our main lower bound will make use of the following technical lemma, the full proof of which is provided in the appendix (Section C.1). Informally, the Lemma states that the surplus enjoyed by an optimal buyer algorithm would only increase if this surplus were viewed without discounting.

**Lemma 4.** *Let the buyer's discount sequence $\{\gamma_t\}$ be positive and nonincreasing. For any seller algorithm $\mathcal{A}$, value distribution $\mathcal{D}$, and surplus-maximizing buyer algorithm $\mathcal{B}^*(\mathcal{A}, \mathcal{D})$,*
$$E\left[\sum_{t=1}^T \gamma_t a_t(v_t - p_t)\right] \leq E\left[\sum_{t=1}^T a_t(v_t - p_t)\right]$$

Notice if $a_t(v_t - p_t) \geq 0$ for all $t$, then the Lemma 4 is trivial. This would occur if the buyer only ever accepts prices less than its value ($a_t = 1$ only if $p_t \leq v_t$). However, Lemma 4 is interesting in that it holds for *any* seller algorithm $\mathcal{A}$. It's easy to imagine a seller algorithm that incentivizes the buyer to sometimes accept a price $p_t > v_t$ with the promise that this will generate better prices in the future (e.g. setting $p_{t'} = 1$ and offering $p_t = 0$ for all $t > t'$ only if $a_{t'} = 1$ and otherwise setting $p_t = 1$ for all $t > t'$).

Lemmas 3 and 4 let us prove our main lower bound.

**Theorem 3.** *Fix a positive, nonincreasing, discount sequence $\{\gamma_t\}$. Let $\mathcal{A}$ be any seller algorithm for the repeated setting. There exists a buyer value distribution $\mathcal{D}$ such that $\text{Regret}(\mathcal{A}, \mathcal{D}, T) \geq \frac{1}{12}T_\gamma$. In particular, if $T_\gamma = \Omega(T)$, no-regret is impossible.*

*Proof.* Let $\{a_{b,t}, p_{b,t}\}$ be the sequence of prices and allocations generated by playing $\mathcal{B}^*(\mathcal{A}, b)$ against $\mathcal{A}$. For each $b \in [0, 1]$ and $p \in [0, 1) \times \{0, 1\}$, let $\mu_b(p, a) = \frac{1}{T_\gamma}\sum_{t=1}^T \gamma_t \mathbf{1}\{a_{b,t} = a\}\mathbf{1}\{p_{b,t} = p\}$. Notice that $\mu_b(p, a) > 0$ for countably many $(p, a)$ and let $\Omega_b = \{(p, a) \in [0, 1] \times \{0, 1\} : \mu_b(p, a) > 0\}$. We think of $\mu_b$ as being a distribution. It's in fact a random measure since the $\{a_{b,t}, p_{b,t}\}$ are themselves random. One could imagine generating $\mu_b$ by playing $\mathcal{B}^*(\mathcal{A}, b)$ against $\mathcal{A}$ and observing the sequence $\{a_{b,t}, p_{b,t}\}$. Every time we observe a price $p_{b,t} = p$ and allocation $a_{b,t} = a$, we assign $\frac{1}{T_\gamma}\gamma_t$ additional mass to $(p, a)$ in $\mu_b$. This is impossible in practice, but the random measure $\mu_b$ has a well-defined distribution.

Now consider the following strategy $\mathcal{S}$ for the single-shot setting. $\mathcal{S}_b$ is induced by drawing a $\mu_b$, then drawing $(p, a) \sim \mu_b$. Note that for any $b \in [0, 1]$ and any measurable function $f$

$$E_{(p,a) \sim S_b}[f(a,p)] = E_{\mu_b \sim S_b}\left[E_{(p,a) \sim \mu_b}[f(a,b) \mid \mu_b]\right] = \tfrac{1}{T_\gamma} E\left[\sum_{t=1}^{T} \gamma_t f(a_{b,t}, p_{b,t})\right].$$

Thus the strategy $\mathcal{S}$ is incentive compatible, since for any $b, v \in [0,1]$

$$E_{(p,a) \sim S_b}[a(v-p)] = \frac{1}{T_\gamma} E\left[\sum_{t=1}^{T} \gamma_t a_{b,t}(v - p_{b,t})\right] = \frac{1}{T_\gamma} \text{BuyerSurplus}(\mathcal{A}, \mathcal{B}^*(\mathcal{A}, b), v, T)$$

$$\leq \frac{1}{T_\gamma} \text{BuyerSurplus}(\mathcal{A}, \mathcal{B}^*(\mathcal{A}, v), v, T) = \frac{1}{T_\gamma} E\left[\sum_{t=1}^{T} \gamma_t a_{v,t}(v - p_{v,t})\right] = E_{(p,a) \sim S_v}[a(v-p)]$$

where the inequality follows from the fact that $\mathcal{B}^*(\mathcal{A}, v)$ is a surplus-maximizing algorithm for a buyer whose value is $v$. The strategy $\mathcal{S}$ is also rational, since for any $v \in [0,1]$

$$E_{(p,a) \sim S_v}[a(v-p)] = \frac{1}{T_\gamma} E\left[\sum_{t=1}^{T} \gamma_t a_{v,t}(v - p_{v,t})\right] = \frac{1}{T_\gamma} \text{BuyerSurplus}(\mathcal{A}, \mathcal{B}^*(\mathcal{A}, v), v, T) \geq 0$$

where the inequality follows from the fact that a surplus-maximizing buyer algorithm cannot earn negative surplus, as a buyer can always reject every price and earn zero surplus.

Let $r_t = 1 - \gamma_t$ and $T_r = \sum_{t=1}^{T} r_t$. Note that $r_t \geq 0$. We have the following for any $v \in [0,1]$:

$$T_\gamma \text{SSRegret}(\mathcal{S}, v) = T_\gamma\left(v - E_{(p,a) \sim S_v}[ap]\right) = T_\gamma\left(v - \frac{1}{T_\gamma} E\left[\sum_{t=1}^{T} \gamma_t a_{v,t} p_{v,t}\right]\right)$$

$$= T_\gamma v - E\left[\sum_{t=1}^{T} \gamma_t a_{v,t} p_{v,t}\right] = (T - T_r)v - E\left[\sum_{t=1}^{T} (1 - r_t) a_{v,t} p_{v,t}\right]$$

$$= Tv - E\left[\sum_{t=1}^{T} a_{v,t} p_{v,t}\right] + E\left[\sum_{t=1}^{T} r_t a_{v,t} p_{v,t}\right] - T_r v$$

$$= \text{Regret}(\mathcal{A}, v, T) + E\left[\sum_{t=1}^{T} r_t a_{v,t} p_{v,t}\right] - T_r v = \text{Regret}(\mathcal{A}, v, T) + E\left[\sum_{t=1}^{T} r_t(a_{v,t} p_{v,t} - v)\right]$$

A closer look at the quantity $E\left[\sum_{t=1}^{T} r_t(a_{v,t} p_{v,t} - v)\right]$, tells us that: $E\left[\sum_{t=1}^{T} r_t(a_{v,t} p_{v,t} - v)\right] \leq E\left[\sum_{t=1}^{T} r_t a_{v,t}(p_{v,t} - v)\right] = -E\left[\sum_{t=1}^{T} (1 - \gamma_t) a_{v,t}(v - p_{v,t})\right] \leq 0$, where the last inequality follows from Lemma 4. Therefore $T_\gamma \text{SSRegret}(\mathcal{S}, v) \leq \text{Regret}(\mathcal{A}, v, T)$ and taking $\mathcal{D}$ to be the point-mass on the value $v \in [0,1]$ which realizes Lemma 3 proves the statement of the theorem. $\quad\square$

# 7 Conclusion

In this work, we have analyzed the performance of revenue maximizing algorithms in the setting of a repeated posted-price auction with a *strategic* buyer. We show that if the buyer values inventory in the present more than in the far future, no-regret (with respect to revenue gained against a truthful buyer) learning is possible. Furthermore, we provide lower bounds that show such an assumption is in fact necessary. These are the first bounds of this type for the presented setting. Future directions of study include studying buyer behavior under weaker polynomial discounting rates as well understanding when existing "off-the-shelf" bandit-algorithm (UCB, or EXP3), perhaps with slight modifications, are able to perform well against strategic buyers.

### Acknowledgements

We thank Corinna Cortes, Gagan Goel, Yishay Mansour, Hamid Nazerzadeh and Noam Nisan for early comments on this work and pointers to relevent literature.

## Footnotes

[1] If there are multiple optimal sequences, the buyer can then choose to randomize over the set of sequences. In such a case, the worst case distribution (for the seller) is the one that always selects the revenue minimizing optimal sequence. In that case, let $a_{1:T}^*$ denote the revenue-minimizing buyer-optimal sequence.

[2]This subclass of auctions is even *ex post* rational.

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
