[Supplementary Material]

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

# A  Upper Bound on the Regret of `Monotone`

## A.1  Proof of Lemma 1

*Proof.* For any sequence $\mathbf{a} \in \{0,1\}^T$ let $\text{last}(\mathbf{a})$ be the last round $t$ where $a_t = 1$ and $a_{t+1} = 0$, or $\text{last}(\mathbf{a}) = 0$ if there is no such round. Let $\mathbf{a}^* = a_1^*, \ldots, a_T^*$, and assume for contradiction that $\text{last}(\mathbf{a}^*) > 0$. Further, assume without loss of generality that $\text{last}(\mathbf{a}^*) \geq \text{last}(\tilde{\mathbf{a}}^*)$ for every optimal sequence $\tilde{\mathbf{a}}^*$. Let $\ell = \text{last}(\mathbf{a}^*)$.

Suppose that $a_t^* = 0$ for all $t \geq \ell + 1$. If $v - p_\ell \geq 0$ then, since $p_{\ell+1} = p_\ell$, letting $a_{\ell+1}^* = 1$ does not decrease the buyer's total surplus and increases $\text{last}(\mathbf{a}^*)$, violating the assumption that $\text{last}(\mathbf{a}^*) \geq \text{last}(\tilde{\mathbf{a}}^*)$ for every optimal sequence $\tilde{\mathbf{a}}^*$. On the other hand, if $v - p_\ell < 0$ then letting $a_\ell^* = 0$ increases the buyer's total surplus, contradicting the optimality of $\mathbf{a}^*$.

Otherwise choose the smallest $k \geq 1$ such that $a_{\ell+k}^* = 0$, and $a_{\ell+k+1}^* = 1$. Note that $p_{\ell+k+1} = \beta^k p_\ell$ and $p_{\ell+k} = \beta^{\ell-1} p_\ell$. Swapping the values of $a_\ell^*$ and $a_{\ell+1}^*$ does not affect the buyer's surplus in rounds other than $\ell$ and $\ell + 1$, and must not increase the buyer's total surplus, which implies $\gamma^{\ell-1}(v - p_\ell) \geq \gamma^\ell(v - \beta p_\ell)$. Likewise, swapping the values of $a_{\ell+k}^*$ and $a_{\ell+k+1}^*$ does not affect the buyer's surplus in rounds other than $\ell + k$ and $\ell + k + 1$, and increases $\text{last}(\mathbf{a}^*)$, so it must decrease the buyer's total surplus, which implies $\gamma^{\ell+k}(v - p_{\ell+k+1}) > \gamma^{\ell+k-1}(v - p_{\ell+k})$.

Cancelling $\gamma$'s in each inequality, and substituting for $p_{\ell+k}$ and $p_{\ell+k+1}$ gives the following inequalities:

$$v - p_\ell \geq \gamma v - \gamma \beta p_\ell \quad \text{and} \quad \gamma v - \gamma \beta^k p_\ell > v - \beta^{k-1} p_\ell$$

Adding the two inequalities and rearranging gives us:

$$\beta^{k-1} p_\ell + \gamma p_\ell (\beta - \beta^k) > p_\ell$$

Dividing through by $p_\ell$ gives us:

$$\beta^{k-1} + \gamma(\beta - \beta^k) > 1 \tag{1}$$

Let $g(\beta) = \beta^{k-1} + \beta - \beta^k$. Since $\beta - \beta^k$ is non-negative and $\gamma \leq 1$, $g(\beta)$ is an upper bound on the left hand side of equation 1. Giving:

$$\beta^{k-1} + \gamma(\beta - \beta^k) \leq g(\beta) \tag{2}$$

However, $\frac{dg}{d\beta} = (k-1)\beta^{k-2} + 1 - k\beta^{k-1} = (1 - \beta^{k-2}) + k(\beta^{k-2} - \beta^{k-1})$, which is non-negative for any $\beta < 1$. To see why, note that both terms in the last expression are non-negative when $k > 1$ and the entire expression is 0 when $k = 1$.

Therefore, $g(\cdot)$ is a non-decreasing function and for any $\beta < 1$, $g(\beta) \leq g(1) = 1$. This fact combined with Eq. (1) and Eq. (2) imply a contradiction. $\qquad\square$

## A.2  Proof of Lemma 2

*Proof.* Rearranging the inequality $t > d_{\beta,\gamma}$ yields $\beta^t (1 + (1 - \beta)T_\gamma) < v$. Subtracting $\beta^{t+1}$ from both sides, multiplying both sides by $\gamma^t$, and applying the inequality $\sum_{t'=1}^{T-t} \gamma^{t'-1} \leq \sum_{t'=1}^{T} \gamma^{t'-1} = T_\gamma$ gives us

$$\gamma^t \left( \beta^t \left( 1 + (1 - \beta) \sum_{t'=1}^{T-t-1} \gamma^{t'-1} \right) - \beta^{t+1} \right) < \gamma^t (v - \beta^{t+1})$$

$$\Leftrightarrow \quad \beta^t (1 - \beta) \sum_{t'=t+1}^{T} \gamma^{t'-1} < \gamma^t (v - \beta^{t+1})$$

Now substitute $\beta^t (1 - \beta) = (v - \beta^{t+1}) - (v - \beta^t)$ and gather terms. We have

$$\sum_{t'=t+2}^{T} \gamma^{t'-1}(v - \beta^{t+1}) < \sum_{t'=t+1}^{T} \gamma^{t'-1}(v - \beta^t). \tag{3}$$

Note that $\sum_{t'=t+1}^{T} \gamma^{t'-1}(v - \beta^t)$ is the surplus of a monotonic buyer that starts accepting (and thus continues to accept) the price offered at time $t+1$. The inequality above, which holds for arbitrary $t > d_{\beta,\gamma}$, states that the surplus that is gained from starting to accept at round $t+1$ is greater than the surplus gained from starting to accept at round $t+2$. Thus, it must be the case $a_{t+1}^* = 1$. $\quad\square$

### A.3 Proof of Corollary 1

Before showing the proof to Corollary 1, we prove the following technical lemma.

**Lemma 5.** $x \geq \log(1+x)$ if $x \geq 0$ and $x \leq 2\log(1+x)$ if $0 \leq x \leq 1$.

*Proof.* By Taylor's theorem $e^x = \sum_{i=0}^{\infty} \frac{x^i}{i!}$. Therefore $e^x \geq 1 + x$ if $x \geq 0$, and so $x \geq \log(1+x)$ if $x \geq 0$. Now let $a_n = \sum_{i=1}^{n}(-1)^{i+1}\frac{x^i}{i}$ and observe that for any positive even integer $n$

$$2a_n = 2x - x^2 + 2\sum_{i=3}^{n}(-1)^{i+1}\frac{x^i}{i}$$

$$= x + (x - x^2) + 2\sum_{i=3,5,7,\dots}^{n} x^i \left(\frac{1}{i} - \frac{x}{i+1}\right)$$

$$\geq x$$

where the inequality follows because $x - x^2 \geq 0$ if $0 \leq x \leq 1$ and $\frac{1}{i} - \frac{x}{i+1} \geq 0$ if $x \leq 1$ and $i \geq 1$. Since $\lim_{n\to\infty} a_n = \log(1+x)$ (by Taylor's theorem) and $\lim_{n\to\infty} a_n = \lim_{n\to\infty, n \text{ even}} a_n$ (because all subsequences of a convergent sequence have the same limit), we have shown $2\log(1+x) \geq x$ for $0 \leq x \leq 1$. $\quad\square$

Now, the proof of Corollary 1.

*Proof of Corollary 1.* From the expression for $\beta$ we have

$$c_{\beta,\gamma} = 1 + \left(1 - \frac{\sqrt{T}}{1+\sqrt{T}}\right)T_\gamma = 1 + \frac{1}{\left(1+\sqrt{T}\right)}T_\gamma = \frac{1 + \sqrt{T} + T_\gamma}{\left(1+\sqrt{T}\right)} \tag{4}$$

which implies

$$1 - \frac{\beta}{c_{\beta,\gamma}} = 1 - \frac{\sqrt{T}}{1+\sqrt{T}+T_\gamma} = \frac{1+T_\gamma}{1+\sqrt{T}+T_\gamma}.$$

We also have

$$d_{\beta,\gamma} = \frac{\log\left(\left(1 + \frac{T_\gamma}{(1+\sqrt{T})}\right)\frac{1}{v}\right)}{\log\left(\frac{1+\sqrt{T}}{\sqrt{T}}\right)} = \frac{\log\left(1 + \frac{T_\gamma}{(1+\sqrt{T})}\right) + \log\left(\frac{1}{v}\right)}{\log\left(1 + \frac{1}{\sqrt{T}}\right)}.$$

By Lemma 5 we know that $x \geq \log(1+x)$ if $x \geq 0$ and $x \leq 2\log(1+x)$ if $0 \leq x \leq 1$. Since $T \geq 1$ we have $\frac{T_\gamma}{(1+\sqrt{T})} \geq 0$ and $0 \leq \frac{1}{\sqrt{T}} \leq 1$ and therefore

$$d_{\beta,\gamma} \leq \frac{2T_\gamma\sqrt{T}}{\left(1+\sqrt{T}\right)} + 2\sqrt{T}\log\left(\frac{1}{v}\right) \leq 2T_\gamma + 2\sqrt{T}\log\left(\frac{1}{v}\right). \tag{5}$$

From the expression for $c_{\beta,\gamma}$ in Eq. (4) we have $\frac{1}{c_{\beta,\gamma}} \leq 1$. Therefore

$$\frac{d_{\beta,\gamma}}{c_{\beta,\gamma}} \leq 2T_\gamma + 2\sqrt{T}\log\left(\frac{1}{v}\right).$$

Now plug the bounds on $1 - \frac{\beta}{c_{\beta,\gamma}}$, $\frac{d_{\beta,\gamma}}{c_{\beta,\gamma}}$ and $\frac{1}{c_{\beta,\gamma}}$ from above into the upper bound from Theorem 1. Noting that $\beta \leq 1$ gives us

$$\text{Regret}(\texttt{Monotone}, v, T) \leq vT\left(\frac{1+T_\gamma}{1+\sqrt{T}+T_\gamma}\right) + v\beta\left(2T_\gamma + 2\sqrt{T}\log\left(\frac{1}{v}\right) + 1\right)$$

$$\leq \sqrt{T}\left(4vT_\gamma + 2v\log\left(\frac{1}{v}\right)\right) + v. \qquad \square$$

## B   Upper Bound on Regret of `Phased`

Let $\lambda$ be a fixed positive constant, whose exact value will be specified later. Define $V_{p,i}^+$ to be the number of explore rounds in phase $i$ where price $p$ was offered and the buyer's value in the round was at least $p + \lambda$. Let $\hat{r}_{p,i}^+ = p\frac{V_{p,i}^+}{S_i}$, and note that $E[\hat{r}_{p,i}^+(\lambda)] = pF(p + \lambda)$. Similarly, define $V_{p,i}^-$ to be the number of explore rounds in phase $i$ where price $p$ was offered and the buyer's value in the round was at least $p - \lambda$. Let $\hat{r}_{p,i}^- = p\frac{V_{p,i}^-}{S_i}$, and note that $E[\hat{r}_{p,i}^-] = pF(p - \lambda)$. Also, let $\tilde{r}_{p,i} = p\frac{A_{p,i}}{S_i}$ be the *observed* revenue of price $p$ in explore rounds in phase $i$.

In the `Phased` algorithm, the price $\tilde{p}_i$ that maximizes $\tilde{r}_{p,i}$ is offered in every exploit round of phase $i$. So our strategy for proving Theorem 2 will be to show that $p^* = \arg\max_p \tilde{r}_{p,i}$ with high probability for all sufficiently large $i$. There are essentially only two ways this can fail to happen: Either the realized buyer values differ greatly from their expectations, or the buyer is untruthful about her realized values. The first case is unlikely, and the latter case is costly to the buyer, provided the number of explore rounds in the phase is sufficiently large. We now quantify 'sufficiently large'. Let $i_p^*$ be the smallest nonnegative integer such that $S_i \geq D_T$ for all $i \geq i_p^*$, where

$$D_T = \max\left(\frac{16}{\Delta^2}\log T, \ \frac{8}{\Delta}C_{\frac{1}{T}}\right)$$

and $C_\delta = T_\gamma(\log(1/\delta) + \log(1/\lambda))$. Note that $i_p^*$ is well-defined because $S_i$ is increasing in $i$. The next lemma uses a standard concentration inequality to bound the probability that certain random variables are close to their expectations.

**Lemma 6.** *Fix price $p \in P$ and phase $i \geq i_p^*$. With probability $1 - 2T^{-1}$*

$$\hat{r}_{p,i}^- \leq pF(p - \lambda) + \frac{\Delta}{4} \quad \text{and} \quad \hat{r}_{p^*,i}^+ \geq p^*F(p^* + \lambda) - \frac{\Delta}{4}.$$

*Proof.* Note that $\hat{r}_{p,i}^-$ is an average of $S_i$ independent random variables, since the variables $p_t$ are chosen deterministically during the explore phase and each $v_t$ is always drawn independently. Also note that $E[\hat{r}_{p,i}^-] = pF(p - \lambda)$. Since $i \geq i_p^*$ we have

$$S_i \geq \frac{16}{\Delta^2}\log T = \frac{1}{(\Delta/4)^2}\log T.$$

Thus by Hoeffding's inequality $\Pr\left[\hat{r}_{p,i}^- \leq pF(p - \lambda) + \frac{\Delta}{4}\right] \geq 1 - T^{-1}$. Similarly $\hat{r}_{p^*,i}^+$ is an average of $S_i$ independent random variables and $E[\hat{r}_{p^*,i}^+] = p^*F(p^* + \lambda)$, and thus $\Pr\left[\hat{r}_{p^*,i}^+ \geq p^*F(p^* + \lambda) - \frac{\Delta}{4}\right] \geq 1 - T^{-1}$. The lemma follows from the union bound. $\square$

Let $\mathcal{L}_{p,i}$ be the set of explore rounds in phase $i$ where the seller offered price $p$ and the buyer $\lambda$-*lied*, i.e., a round $t$ where either the buyer accepted price $p$ and her value $v_t \leq p - \lambda$, or rejected price $p$ and her value $v_t > p + \lambda$. Let $L_{p,i} = |\mathcal{L}_{p,i}|$. The next lemma shows that, for any phase $i$ where the event from the previous lemma occurs, if the observed revenue of the optimal price $p^*$ is less than the observed revenue of another price then the buyer must have told many $\lambda$-lies during phase $i$.

**Lemma 7.** *Fix price $p \in P$ and phase $i$. If $\tilde{r}_{p^*,i} < \tilde{r}_{p,i}$ and the event from Lemma 6 occurs then $L_{p,i} \geq \left(\frac{\Delta - 8K\lambda}{4p}\right)S_i$ or $L_{p^*,i} \geq \left(\frac{\Delta - 8K\lambda}{4p^*}\right)S_i$.*

*Proof.* Assume for contradiction that $L_{p,i} < \left(\frac{\Delta - 8K\lambda}{4p}\right)S_i$ and $L_{p^*,i} < \left(\frac{\Delta - 8K\lambda}{4p^*}\right)S_i$. For any price $p'$ note that $A_{p',i} - V_{p',i}^- \leq L_{p',i}$ and $V_{p',i}^+ - A_{p',i} \leq L_{p',i}$, since $A_{p',i}$ counts the number of times the buyer accepted price $p'$ in phase $i$. Combining these bounds and applying the definitions of $\tilde{r}_{p,i}, \tilde{r}_{p^*,i}, \hat{r}_{p,i}^-$ and $\hat{r}_{p^*,i}^+$ proves

$$\tilde{r}_{p,i} - \hat{r}_{p,i}^- < \frac{\Delta}{4} - K\lambda, \qquad (6)$$

$$\hat{r}^+_{p^*,i} - \tilde{r}_{p^*,i} < \frac{\Delta}{4} - K\lambda. \tag{7}$$

Now observe

$$\tilde{r}_{p,i} < \hat{r}^-_{p,i} + \frac{\Delta}{4} - K\lambda \qquad\qquad \text{Eq. (6)}$$

$$\leq pF(p-\lambda) + \frac{\Delta}{2} - K\lambda \qquad\qquad \text{Lemma 6}$$

$$\leq pF(p) + \frac{\Delta}{2} \qquad\qquad K\text{-Lipschitz continuity}$$

$$\leq p^*F(p^*) - \frac{\Delta}{2} \qquad\qquad \text{Definition of } \Delta$$

$$\leq p^*F(p^* + \lambda) - \frac{\Delta}{2} + K\lambda \qquad\qquad K\text{-Lipschitz continuity}$$

$$\leq \hat{r}^+_{p^*,i} - \frac{\Delta}{4} + K\lambda \qquad\qquad \text{Lemma 6}$$

$$< \tilde{r}_{p^*,i} \qquad\qquad \text{Eq. (7)}$$

which contradicts $\tilde{r}_{p^*,i} < \tilde{r}_{p,i}$. $\qquad\square$

Next we show that the number of $\lambda$-lies told by a surplus-maximizing buyer in any phase is bounded with high probability. This is the main technical lemma.

**Lemma 8.** *Fix price $p \in \mathcal{P}$, phase $i$, and suppose the buyer uses a surplus-maximizing algorithm $\mathcal{B}^*(\text{Phased}, \mathcal{D})$. For all $\delta > 0$ we have $\Pr[L_{p,i} \geq C_\delta] \leq \delta$.*

*Proof.* Let $\mathcal{B}^i$ be a buyer algorithm that acts according to $\mathcal{B}^*(\text{Phased}, \mathcal{D})$ during the first $i - 1$ phases, and from phase $i$ onwards acts truthfully in every round, i.e., $a_t = \mathbf{1}\{v_t \geq p_t\}$ for all rounds $t$ in phases $i, i+1, \ldots, \lceil \log_2 T \rceil$. Assume $\Pr[L_{p,i} \geq C_\delta] > \delta$. We will show that this implies

$$\text{BuyerSurplus}(\text{Phased}, \mathcal{B}^*(\text{Phased}, \mathcal{D}), \mathcal{D}, T) < \text{BuyerSurplus}(\text{Phased}, \mathcal{B}^i, \mathcal{D}, T),$$

a contradiction.

Let $p_1^*, \ldots, p_T^*$ and $a_1^*, \ldots, a_T^*$ be the prices and accept decisions from all rounds when the buyer algorithm is $\mathcal{B}^*(\text{Phased}, \mathcal{D})$, and let $p_1^i, \ldots, p_T^i$ and $a_1^i, \ldots, a_T^i$ be the price and accept decisions from all rounds when the buyer algorithm is $\mathcal{B}^i$. Recall that the values $v_1, \ldots, v_T$ are drawn independently of seller or buyer behavior. Let $t_i^-$ and $t_i^+$ be the first and last explore rounds in phase $i$, respectively. We have

$$\text{BuyerSurplus}(\text{Phased}, \mathcal{B}^*(\text{Phased}, \mathcal{D}), \mathcal{D}, T) - \text{BuyerSurplus}(\text{Phased}, \mathcal{B}^i, \mathcal{D}, T)$$

$$= E\left[\sum_{t=1}^{t_i^- - 1} \gamma^{t-1}(a_t^*(v_t - p_t^*) - a_t^i(v_t - p_t^i))\right] + E\left[\sum_{t=t_i^-}^{t_i^+} \gamma^{t-1}(a_t^*(v_t - p_t^*) - a_t^i(v_t - p_t^i))\right]$$

$$+ E\left[\sum_{t=t_i^+ + 1}^{T} \gamma^{t-1}(a_t^*(v_t - p_t^*) - a_t^i(v_t - p_t^i))\right] \tag{8}$$

$$= E\left[\sum_{t=t_i^-}^{t_i^+} \gamma^{t-1}(a_t^*(v_t - p_t^*) - a_t^i(v_t - p_t^i))\right] + E\left[\sum_{t=t_i^+ + 1}^{T} \gamma^{t-1}(a_t^*(v_t - p_t^*) - a_t^i(v_t - p_t^i))\right] \tag{9}$$

$$= E\left[\sum_{t=t_i^-}^{t_i^+} \gamma^{t-1}(a_t^* - a_t^i)(v_t - p_t^i)\right] + E\left[\sum_{t=t_i^+ + 1}^{T} \gamma^{t-1}(a_t^*(v_t - p_t^*) - a_t^i(v_t - p_t^i))\right] \tag{10}$$

$$\leq E\left[\sum_{t=t_i^-}^{t_i^+} \gamma^{t-1}(a_t^* - a_t^i)(v_t - p_t^i)\right] + \gamma^{t_i^+} T_\gamma \tag{11}$$

$$= \Pr[L_{p,i} \geq C_\delta] E\left[\sum_{t=t_i^-}^{t_i^+} \gamma^{t-1}(a_t^* - a_t^i)(v_t - p_t^i) \mid L_{p,i} \geq C_\delta\right]$$

$$+ \Pr[L_{p,i} < C_\delta] E\left[\sum_{t=t_i^-}^{t_i^+} \gamma^{t-1}(a_t^* - a_t^i)(v_t - p_t^i) \mid L_{p,i} < C_\delta\right] + \gamma^{t_i^+} T_\gamma$$

$$\leq \Pr[L_{p,i} \geq C_\delta] E\left[\sum_{t \in \mathcal{L}_{p,i}} \gamma^{t-1}(a_t^* - a_t^i)(v_t - p_t^i) \mid L_{p,i} \geq C_\delta\right] + \gamma^{t_i^+} T_\gamma \tag{12}$$

$$\leq \Pr[L_{p,i} \geq C_\delta] E\left[\sum_{t \in \mathcal{L}_{p,i}} \gamma^{t-1}(-\lambda) \mid L_{p,i} \geq C_\delta\right] + \gamma^{t_i^+} T_\gamma \tag{13}$$

$$\leq \Pr[L_{p,i} \geq C_\delta] \sum_{t=t_i^+-C_\delta+1}^{t_i^+} \gamma^{t-1}(-\lambda) + \gamma^{t_i^+} T_\gamma \tag{14}$$

$$< \delta \sum_{t=t_i^+-C_\delta+1}^{t_i^+} \gamma^{t-1}(-\lambda) + \gamma^{t_i^+} T_\gamma \tag{15}$$

$$= -\delta\lambda\gamma^{t_i^+-C_\delta}\left(\frac{1-\gamma^{C_\delta}}{1-\gamma}\right) + \gamma^{t_i^+} T_\gamma \leq \frac{\gamma^{t_i^+}}{1-\gamma}\left(\frac{-\delta\lambda}{\gamma^{C_\delta}} + (1-\gamma)T_\gamma\right) \leq 0 \tag{16}$$

Eq. (8) follows from the definition of surplus and the linearity of expectation. Eq. (9) holds because $\mathcal{B}^*(\texttt{Phased}, \mathcal{D})$ and $\mathcal{B}^i$ behave identically before phase $i$. Eq. (10) holds because the prices offered during explore rounds are independent of the buyer's algorithm, and thus $p_t^i = p_t^*$ for $t \in \{t_i^-, \ldots, t_i^+\}$. The fact that $a_t^i = \mathbf{1}\{v_t \geq p_t^i\}$ for $t \geq t_i^-$ implies $a_t^*(v_t - p_t^*) - a_t^i(v_t - p_t^i) \leq 1$ for $t \geq t_i^-$, which yields Eq. (11), and also implies $(a_t^* - a_t^i)(v_t - p_t^i) \leq 0$ for $t \geq t_i^-$, which yields Eq. (12) (recall that $\mathcal{L}_{p,i} \subseteq \{t_i^-, \ldots, t_i^+\}$). The definition of $\lambda$-lies and the fact that $p_t^i = p_t^*$ for $t \in \mathcal{L}_{p,i}$ implies Eq. (13). Eq. (14) holds because $\gamma^{t-1}$ is decreasing in $t$. Eq. (15) follows from our assumption that $\Pr[L_{p,i} \geq C_\delta] > \delta$. Eq. (16) follows from the definition of $C_\delta$. $\qquad\square$

We are ready to prove an upper bound on the regret of the Phased algorithm.

*Proof of Theorem 2.* Let $\mathcal{T}_i^{\text{explore}}$ and $\mathcal{T}_i^{\text{exploit}}$ be the set of explore and exploit rounds of phase $i$. Note that for the Phased algorithm the behavior of a buyer during exploit rounds does not affect the prices offered in future rounds. Since $\tilde{p}_i$ is the price offered in each exploit round of phase $i$, a surplus-maximizing buyer will choose $a_t = \mathbf{1}\{v_t \geq \tilde{p}_i\}$ in any exploit round $t$ of phase $i$. So we can upper bound the regret of the Phased algorithm in terms of the number of explore rounds and the probability that $\tilde{p}_i \neq p^*$ during exploit rounds. We have

$$\text{Regret}(\texttt{Phased}, \mathcal{D}, T) = E\left[\sum_{t=1}^{T} p^* F(p^*) - a_t p_t\right]$$

$$= \sum_i \sum_{t \in \mathcal{T}_i^{\text{explore}}} E\left[p^* F(p^*) - a_t p_t\right] + \sum_i \sum_{t \in \mathcal{T}_i^{\text{exploit}}} E\left[p^* F(p^*) - a_t p_t\right]$$

$$\leq \sum_i |\mathcal{P}| S_i + \sum_i \sum_{p \in \mathcal{P}\backslash\{p^*\}} \Pr\left[\tilde{p}_i = p\right] (T_i - |\mathcal{P}| S_i)$$

$$\leq \sum_i |\mathcal{P}| S_i + \sum_{p \in \mathcal{P}\backslash\{p^*\}} \sum_{i \leq i_p^*} T_i + \sum_{p \in \mathcal{P}\backslash\{p^*\}} \sum_{i > i_p^*} \Pr\left[\tilde{p}_i = p\right] T_i \tag{17}$$

where expectations and probabilities are with respect to value distribution $\mathcal{D}$, seller algorithm Phased, and buyer algorithm $\mathcal{B}^*(\text{Phased}, \mathcal{D})$. We will now bound each term in Eq. (17). Let $\lambda = \frac{\Delta}{16K}$.

Let $n = \lceil \log_2 T \rceil$. Recall that $T_i = 2^i$ and $S_i = T_i^\alpha$, which implies $\sum_i S_i = \sum_{i=1}^{n-1} 2^{\alpha i}$. Since $n \leq \log_2 T + 1$ we have $2^n \leq 2T$. Thus

$$\sum_i S_i = \sum_{i=1}^{n-1} 2^{\alpha i} \leq \frac{(2^\alpha)^n - 1}{2^\alpha - 1} = \frac{(2^n)^\alpha - 1}{2^\alpha - 1} \leq \frac{2^\alpha T^\alpha - 1}{2^\alpha - 1} \leq \frac{2^\alpha}{2^\alpha - 1} T^\alpha. \tag{18}$$

where the first inequality follows from the formula for a geometric series (this is just the standard 'doubling trick').

By the definition of $S_i$ and $i_p^*$ we have $T_{i_p^* - 1} < (D_T)^{1/\alpha}$, which implies $T_{i_p^* + 1} \leq 4(D_T)^{1/\alpha}$. Also note that $\sum_{j \leq i} T_j \leq T_{i+1}$ for all $i$, again because $T_i = 2^i$. Thus

$$\sum_{p \in \mathcal{P} \setminus \{p^*\}} \sum_{i \leq i_p^*} T_i \leq \sum_{p \in \mathcal{P} \setminus \{p^*\}} 4(D_T)^{1/\alpha} \tag{19}$$

Finally, for any $p \neq p^*$ and $i > i_p^*$ if $\tilde{p}_i = p$ then $\tilde{r}_{p^*, i} < \tilde{r}_{p, i}$, which by Lemma 7 implies that either the event from Lemma 6 does not occur,

$$L_{p, i} \geq \frac{\Delta - 8K\lambda}{4p} S_i, \text{ or} \tag{20}$$

$$L_{p^*, i} \geq \frac{\Delta - 8K\lambda}{4p^*} S_i. \tag{21}$$

Since $\lambda = \frac{\Delta}{16K}$ and $p, p^* < 1$, Eq. (20) and Eq. (21) respectively imply

$$L_{p, i} \geq \frac{\Delta}{8} S_i, \text{ or} \tag{22}$$

$$L_{p^*, i} \geq \frac{\Delta}{8} S_i. \tag{23}$$

The event from Lemma 6 occurs with probability $1 - 2T^{-1}$. And since $S_i \geq D_T \geq (8/\Delta) C_{\frac{1}{T}}$ for all $i \geq i_p^*$, we have that Eq. (22) and Eq. (23) imply either $L_{p, i} \geq C_{\frac{1}{T}}$ or $L_{p^*, i} \geq C_{\frac{1}{T}}$, which by Lemma 8 each occur with probability at most $T^{-1}$. Thus by the union bound $\Pr[\tilde{p}_i = p] \leq 4T^{-1}$, and therefore

$$\sum_{p \in \mathcal{P} \setminus \{p^*\}} \sum_{i > i_p^*} \Pr[\tilde{p}_i = p] T_i \leq 4|\mathcal{P}| \tag{24}$$

Combining Eqs. (18), (19) and (24) with Eq. (17) yields

$$\text{Regret}(\text{Phased}, \mathcal{D}, T) \leq \frac{2^\alpha}{2^\alpha - 1} |\mathcal{P}| T^\alpha + \sum_{p \in \mathcal{P} \setminus \{p^*\}} 4(D_T)^{1/\alpha} + 4|\mathcal{P}|$$

Plugging in the definition $D_T$ and $\lambda = \frac{\Delta}{16K}$, we have

$$\text{Regret}(\text{Phased}, \mathcal{D}, T) \leq \frac{2^\alpha}{2^\alpha - 1} |\mathcal{P}| T^\alpha + \sum_{p \in \mathcal{P} \setminus \{p^*\}} 4 \left( \frac{16}{\Delta^2} \log T \right)^{1/\alpha}$$

$$+ \sum_{p \in \mathcal{P} \setminus \{p^*\}} 4 \left( \frac{8}{\Delta} T_\gamma (\log T + \log(16K/\Delta)) \right)^{1/\alpha} + 8|\mathcal{P}|.$$

and simplifying yields the statement of the theorem. $\qquad\square$

## C    Lower Bound Proofs

### C.1    Proof of Lemma 3

*Proof.* Fix a incentive compatible and rational strategy $\mathcal{S}$. Let $\text{SellerRevenue}(b) = E_{(p,a) \sim \mathcal{S}_b}[ap]$ be the seller's expected revenue if the buyer bids $b$, and let $\text{BuyerSurplus}(b, v) = E_{(p,a) \sim \mathcal{S}_b}[a(v -$

$p)]$ be the buyer's expected surplus if she bids $b$ and her value is $v$. It suffices to show that there exists $v \in [0, 1]$ such that $v - \text{SellerRevenue}(v) \geq \frac{1}{12}$.

Before proceeding, we establish some properties of $\mathcal{S}$. Incentive compatibility of $\mathcal{S}$ ensures that

$$\text{BuyerSurplus}(v, v) \geq \text{BuyerSurplus}(b, v) \tag{25}$$

for all $b, v \in [0, 1]$, and rationality of $\mathcal{S}$ ensures that

$$\text{BuyerSurplus}(v, v) \geq 0 \tag{26}$$

for all $v \in [0, 1]$. Also

$$\text{SellerRevenue}(b) + \text{BuyerSurplus}(b, v) = E_{(p,a)\sim\mathcal{S}_b}[a]v \tag{27}$$

for all $b, v \in [0, 1]$, which follows directly from definitions, and

$$\text{SellerRevenue}(v) \leq E_{(p,a)\sim\mathcal{S}_v}[a]v \tag{28}$$

for all $v \in [0, 1]$, which follows from rationality: By (27) we have $\text{BuyerSurplus}(v, v) = E_{(p,a)\sim\mathcal{S}_v}[a]v - \text{SellerRevenue}(v)$, and thus if (28) were false we would have $\text{BuyerSurplus}(v, v) < 0$, which contradicts (26).

Now observe that for any $b, v \in [0, 1]$

$$
\begin{aligned}
v - \text{SellerRevenue}(v) &\geq E_{(p,a)\sim\mathcal{S}_v}[a]v - \text{SellerRevenue}(v) \\
&= \text{BuyerSurplus}(v, v) \\
&\geq \text{BuyerSurplus}(b, v) \\
&= E_{(p,a)\sim\mathcal{S}_b}[a(v - p)] \\
&= E_{(p,a)\sim\mathcal{S}_b}[a]v - E_{(p,a)\sim\mathcal{S}_b}[ap] \\
&= E_{(p,a)\sim\mathcal{S}_b}[a]v - \text{SellerRevenue}(b) \\
&\geq \left(\frac{\text{SellerRevenue}(b)}{b}\right)v - \text{SellerRevenue}(b) \\
&= (v - b)\left(\frac{\text{SellerRevenue}(b)}{b}\right)
\end{aligned}
$$

(29) and (30) and (31) labels on respective lines.

$$\text{BuyerSurplus}(v, v) \tag{29}$$
$$\text{BuyerSurplus}(b, v) \tag{30}$$
$$\left(\frac{\text{SellerRevenue}(b)}{b}\right)v - \text{SellerRevenue}(b) \tag{31}$$

where (29) follows from (27), (30) follows from (25), and (31) follows from (28). Now let $b = \frac{1}{4}$ and $v = \frac{1}{2}$. If $v - \text{SellerRevenue}(v) \geq \frac{1}{6}$ we are done. Otherwise the first and last lines from the above chain of inequalities and $v - \text{SellerRevenue}(v) < \frac{1}{6}$ imply

$$\frac{\text{SellerRevenue}(b)}{b} \leq \frac{v - \text{SellerRevenue}(v)}{v - b} < \frac{1}{6}\frac{1}{v - b} = \frac{2}{3}$$

which can be rearranged into $b - \text{SellerRevenue}(b) \geq \frac{1}{3}b \geq \frac{1}{12}$. □

### C.2 Proof of Lemma 4

*Proof.* It will be convenient to define the following (all expectations in these definitions are with respect to $\mathcal{A}, \mathcal{D}$ and $\mathcal{B}^*(\mathcal{A}, \mathcal{D})$):

$$\text{rev}(t_1, t_2) = E\left[\sum_{t=t_1}^{t_2} a_t p_t\right]$$

$$\text{sur}(t_1, t_2) = E\left[\sum_{t=t_1}^{t_2} \gamma_t a_t(v_t - p_t)\right]$$

$$\text{udsur}(t_1, t_2) = E\left[\sum_{t=t_1}^{t_2} a_t(v_t - p_t)\right]$$

$$\text{totval}(t_1, t_2) = E\left[\sum_{t=t_1}^{t_2} a_t v_t\right]$$

where "udsur" stands for "undiscounted surplus" and "totval" stands for "total value". Note that by definition

$$\text{rev}(t_1, t_2) + \text{udsur}(t_1, t_2) = \text{totval}(t_1, t_2). \tag{32}$$

Also, since $\mathcal{B}^*(\mathcal{A}, \mathcal{D})$ is a surplus-maximizing buyer strategy, $\text{sur}(t, T) \geq 0$ for all rounds $t$, because otherwise the buyer could increase her surplus by following $\mathcal{B}^*(\mathcal{A}, \mathcal{D})$ until round $t - 1$ and then selecting $a_{t'} = 0$ for all rounds $t' \geq t$.

We will first prove that $\text{sur}(t, T) \leq \gamma_t \text{udsur}(t, T)$ for all rounds $t$. The proof will proceed by induction. For the base case, we have $\text{sur}(T, T) = \gamma_T \text{udsur}(T, T)$ by definition. Now assume for the inductive hypothesis that $\text{sur}(t + 1, T) \leq \gamma_{t+1} \text{udsur}(t + 1, T)$. Since $\text{sur}(t + 1, T) \geq 0$ and $\gamma_{t+1} > 0$, by the inductive hypothesis we have $\text{udsur}(t + 1, T) \geq 0$. Therefore

$$
\begin{aligned}
\text{sur}(t, T) &= \text{sur}(t, t) + \text{sur}(t + 1, T) \\
&= \gamma_t \text{udsur}(t, t) + \text{sur}(t + 1, T) \\
&\leq \gamma_t \text{udsur}(t, t) + \gamma_{t+1} \text{udsur}(t + 1, T) \tag{33} \\
&\leq \gamma_t \text{udsur}(t, t) + \gamma_t \text{udsur}(t + 1, T) \tag{34} \\
&= \gamma_t \text{udsur}(t, T)
\end{aligned}
$$

where Eq. (33) follows from the inductive hypothesis and Eq. (34) follows because $\text{udsur}(t+1, T) \geq 0$ and $\gamma_t \geq \gamma_{t+1}$. Thus $\text{sur}(t, T) \leq \gamma_t \text{udsur}(t, T)$.

Since $\text{sur}(1, T) \leq \gamma_1 \text{udsur}(1, T)$ and $\gamma_1 \leq 1$, by Eq. (32) we have $\text{rev}(1, T) + \text{sur}(1, T) \leq \text{totval}(1, T)$, which proves the lemma. $\square$