[Reviews · NeurIPS 2013]

Submitted by Assigned_Reviewer_3

This paper studies the problem of inferring a buyer’s value for a good when the buyer is repeatedly interacting with the seller through a posted-price mechanism. The buyer is modeled as a strategic agent, interested in maximizing her long-term surplus; the target of this work is to optimize seller's revenue.

Pros:
1. The problem is novel and interesting.
2. Some results are very nice, such as Theorem 3 in Section 6.2.

Cons:
1. My major concern about this work is its setting. It shows that the discount factor added to the buyer is very critical: if the buyer's payoff is not discounted, no regret learning is possible. This result looks reasonable. However, while analyzing the seller's revenue, no discount factor is introduced. Such a setting is a bit strange: there are two players in this game, but only one player is discounted over time horizon. A more natural setting is to discount both.

2. It is better to give some intuitive explanations to the theorems. For example, does Theorem 1 tell that the monotone algorithm is good or not? Similarly, can we claim the phased algorithm is a good one based on Theorem 2?

Minor issues:
1. The monotone algorithm given in line 194-195 is opposite to the explanation of this algorithm in line 196-198. I assume this is a typo.

2. c_{\beta,\gamma) and d_{\beta,\gamma) are not defined in Theorem 1. Similarly, the sequence {a_i^*} is not defined in Lemma 5.


Overall, the paper is very interesting and has the potential to be a high-quality paper.
Summary: Overall, the paper is very interesting and has the potential to be a high-quality paper. It can be improved from several aspects: (1) problem setting, (2) theoretical results interpretation, and (3) paper writing.

Submitted by Assigned_Reviewer_4

This paper asks an interesting question: When a single buyer wishes to buy many instances of the same good from a single seller, in a repeated auction setting, how should the seller behave? The seller presumably would like to learn the buyer's valuation (distribution) over time, but the risk is that a patient buyer may be able to mislead the seller and take advantage of this in future rounds.

Model: The seller is restricted to simply posting a price on each round. The buyer's valuation is drawn fresh on each round from a fixed distribution that's unknown to the seller.

I'm somewhat surprised this question hasn't been asked before, since it seems quite natural, and I suspect there may be related work on repeated games with incomplete information, but a quick search didn't turn up anything strongly relevant.

One nice thing in the paper is the definition of seller regret as the difference between his expected revenue and that he could have earned if he had offered the best fixed price on all rounds *and the buyer had been truthful*.

The results are pretty much what you would expect: a lower bound showing that yes, a patient buyer can take advantage of the seller; and a positive result that, in epochs of exponentially increasing length, tries every possible price a large number of times and then uses for a while the one that does best---he idea is that eventually (due to buyer time-discounting) the epochs will be long enough that lying is not advantageous.

I like that this paper asks an interesting question in the right way, but I'm just not that excited about the results (or the techniques).
Summary: I like that this paper asks an interesting question in the right way, but I'm just not that excited about the results (or the techniques).

Submitted by Assigned_Reviewer_5

This paper explores the problem of online learning near-optimal posted-prices when facing the same strategic buyer in repeated auctions. Assuming the buyer’s discount factor is large enough, when the buyer value is an unknown fixed value, the authors propose an algorithm which monotonically decrease posted-prices and show that the algorithm only suffers sublinear regrets. When the buyer value is drawn from a distribution, the authors propose a multi-phase exploration-exploitation algorithm with no-regret. The exploration is not just for estimating the buyer value, it also discourages buyers to manipulate the algorithm. In the end, the authors also show that if the buyer discount factor is not *large enough*, it’s impossible to have any no-regret algorithms.

In general, I like this paper and think it addresses a very interesting problem. Most other online pricing papers usually assume buyers are myopic or just participate once. It’s nice to see that given some conditions, no-regret algorithms are still possible even if the buyer participates multiple times and are strategic.

Below are some detailed comments.

- In the stochastic value setting, you assume the set of possible prices is finite. Although I think it’s fine to assume finite possible prices (if we consider the minimum unit of currency), the result is still slightly unsatisfying especially your regret bound depends on both the number of possible prices and \Delta_p (\Delta_p would be very small if the prices are close.) That being said, I still like the current result. I just think it would be much nicer to see discussion for infinite or large number of possible prices.

- The analysis is sometimes hard to follow. In particular, some notations are used without being defined (e.g., c_{beta,gamma} and d_{beta,gamma} in Theorem 1.) I know they are defined in the appendix, but it would be helpful to mentioned it in the main text. It would also be helpful to put more intuitions in the main paper. For example, there are some nice proof intuitions in the appendix (e.g., the proof in Appendix B). I think it would help to put some of them in the main text if space allows.
Summary: Overall, I think this is a solid theory paper addressing an interesting problem. I would recommend to accept this paper.
Author Feedback

Author rebuttal: All reviewers seem to agree this is a well-stated and important problem and we thank them for their careful reading of the paper. We believe that this paper provides a significant step towards better understanding this setting and will draw further attention to the problem. Below we address the reviewers' specific concerns.

1) "a more natural setting is to discount both [buyer and seller]."

While this 'double-discounting' setting is more general, our 'single-discounting' setting already has considerable practical applicability. There are many important real-world markets where sellers are far more willing to wait for revenue than buyers are willing to wait for goods, particularly in online advertising. For example, advertisers are often interested in showing ads to users who have recently viewed their products online -- this practice is called 'retargeting' -- and the value of these user impressions decays rapidly over time. Or consider an ad campaign that is tied to a product launch. A user impression that is purchased long after the launch -- such as the release of a movie -- is almost worthless. In all these examples, the seller's discount rate is effectively 1 compared to the buyer's.

Nonetheless, an algorithm for the double discounting setting is an interesting avenue for future research. We would be happy to write this as an open problem in the paper. Again, we emphasize the role of this paper in drawing attention to this problem.

2) "...give some intuitive explanations to the theorems...[are they] good or not?"

We agree that the results deserve a high level summary. We emphasize that our bounds show that the algorithms are `no regret', i.e. their average regret approaches zero as the seller's horizon T increases. Also, our lower bound proves that a buyer discount rate less than 1 is necessary for any algorithm to be no-regret. These are the first bounds of this type for the presented setting. To gain further intuition regarding the bounds, it helps to parametrize the seller horizon T_\gamma as a function of T, e.g. T_\gamma = T^{c} for 0 <= c <= 1. Writing it in this fashion, we see that the Monotone algorithm has regret at most O(T^{c + 1/2}), and the Phased algorithm has regret at most O((log T)^{1/\sqrt{c}} T^{\sqrt{c}}) if we choose \alpha = \sqrt{c}. The lower bound in Corollary 3 states that, in the worst case, any seller will incur a regret of at least Omega(T^c). Closing the gap between these upper and lower bounds will be the subject of further research. We will include this discussion in a camera-ready version.

3) To address some of the more minor concerns:

- "regret bound depends on both the number of possible prices and \Delta_p"

This is a fair point, but please note that these are a standard quantities in stochastic bandit problems. For example, the UCB algorithm's regret bound also depends on these quantities.

- "I suspect there may be related work on repeated games with incomplete information"

We also researched this literature, and were surprised not to find anything particularly relevant. Much previous work has focused on characterizing the equilibria of repeated games. Also note that the repeated game we study is not zero-sum.

- "discussion for infinite or large number of possible prices"

This is a very interesting direction, though somewhat orthogonal to the focus of our submission in the sense that continuous action spaces also warrant special treatment in standard online learning models. We believe techniques from the literature on 'continuum-armed bandits' [Agrawal 96, Kleinberg 2005] could be adapted to our setting.

- All reviewers felt that the proofs could have been improved by moving the exposition from the appendix to the main body. We agree, and will do so. We will also correct the issues with notation.

References:
Agrawal, Rajeev (1995) "The continuum-armed bandit problem". SIAM Journal on Control and Optimization.
Kleinberg, Robert (2005) "Nearly tight bounds for the continuum-armed bandit problem". NIPS 17.